# Desensitization Protocols for Anti-SARS-CoV-2 Vaccines in Patients with High Risk of Allergic Reactions

**DOI:** 10.3390/vaccines11050910

**Published:** 2023-04-27

**Authors:** Maria Rita Messina, Carlotta Crisciotti, Laura Pellegrini, Emanuele Nappi, Francesca Racca, Giovanni Costanzo, Lorenzo Del Moro, Sebastian Ferri, Francesca Puggioni, Giorgio Walter Canonica, Enrico Heffler, Giovanni Paoletti

**Affiliations:** 1Personalized Medicine, Asthma and Allergy, IRCCS Humanitas Research Hospital, Rozzano, 20089 Milan, Italy; mariarita.messina@humanitas.it (M.R.M.); carlotta.crisciotti@st.hunimed.eu (C.C.); laura.pellegrini@st.hunimed.eu (L.P.); emanuele.nappi@humanitas.it (E.N.); francesca.racca@humanitas.it (F.R.); giovanni.costanzo@humanitas.it (G.C.); lorenzo.delmoro@humanitas.it (L.D.M.); sebastian.ferri@humanitas.it (S.F.); franscesca.puggioni@humanitas.it (F.P.); giorgio_walter.canonica@hunimed.eu (G.W.C.); giovanni.paoletti@hunimed.eu (G.P.); 2Department of Clinical and Experimental Medicine, University of Florence, 50139 Florence, Italy; 3Department of Biomedical Sciences, Humanitas University, Pieve Emanuele, 20090 Milan, Italy

**Keywords:** SARS-CoV-2, COVID-19, vaccine, allergy, desensitization

## Abstract

Vaccines for SAR-CoV-2 are the most effective preventive treatment able to reduce the risk of contracting the infection and experiencing worse outcomes whenever the infection is contracted. Despite their rarity, hypersensitivity reactions to the anti-SARS-CoV-2 vaccine have been described and could become the reason not to complete the vaccination. Desensitization protocols for other vaccines have been described and validated, while the use of this approach for anti-SARS-CoV-2 vaccines is still anecdotal. We herein describe our experience with 30 patients with previous allergic reactions to anti-SARS-CoV-2 vaccines or to any of their excipients, proving that they are effective and safe; only two patients experienced hypersensitivity reaction symptoms during the desensitization procedure. Moreover, in this article, we propose desensitization protocols for the most common anti-SARS-CoV-2 vaccines.

## 1. Introduction

For more than three years now, the whole world has been facing the Severe Acute Respiratory Syndrome CoronaVirus 2 (SARS-CoV-2) viral infection pandemic, responsible for CoronaVirus Disease-2019 (COVID-19) for over 760 million confirmed cases of infection and more than 6.8 million deaths worldwide so far [1]. Anti-SARS-CoV-2 vaccination has been the best tool so far available to significantly reduce the impact of the COVID-19 pandemic in terms of both morbidity and mortality. A report by the Italian National Institute of Health estimated the number of infections, hospitalizations and deaths associated with SARS-CoV-2 directly avoided by anti-SARS-CoV-2 vaccination in the period January 2021–September 2021; it found more than 445,000 infections, more than 79,000 hospitalizations in medical units, over 9800 hospitalizations in intensive care units and over 220,000 deaths were averted during the period analyzed [2]. 

International drug agencies and the main Allergy and Clinical Immunology scientific societies currently recommend that, apart from allergy to one of the vaccine components, there are no absolute contraindications to the administration of anti-SARS-CoV-2 vaccines [3,4]. Severe allergic reactions to anti-COVID-19 vaccines are, however, extremely rare. From an analysis of more than 17 million individuals who underwent vaccination in the United States between 14 December 2020, and 16 January 2021, it appeared that anaphylaxis occurred at a rate of 2.5 and 4.9 per million doses of the mRNA-1273 and BNT162b2 vaccine, respectively. Most reactions occurred following the first vaccine dose and in the absence of a previous history of anaphylaxis, remarking the importance of adequate training and equipment for the management of severe allergic reactions in all vaccination centers. Interestingly, nearly all the reported anaphylaxes occurred in female individuals (63 out of 66) [5]. 

A similar frequency of severe allergic reactions was also seen in the Italian population: data from the Italian Drug Agency showed that as of December 2021, following the administration of over 100 million vaccine doses, 299 cases of suspect systemic hypersensitivity reactions occurred, of which only 111 were classified as at high probability of anaphylaxis according to the Brighton Collaboration classification [6,7]. 

An allergological evaluation is recommended for people with suspected hypersensitivity to any component of anti-SARS-CoV-2 vaccines [8,9]. The precise mechanisms underlying hypersensitivity reactions to anti-SARS-CoV-2 vaccines are not yet completely elucidated; thus, diagnostic procedures to predict allergic reactions to anti-SARS-CoV-2 vaccines do not seem to be consistently recommended. On the basis of the data available in the literature and the experience in real life, it is deemed that skin tests with Polyethylene glycol (PEG), Polysorbate 80 and Tromethamine (excipients contained in the current available anti-SARS-CoV-2 vaccines and with a certain degree of known ability to induce allergic reactions [10,11,12]) have a debated role in identifying individuals at risk of developing hypersensitivity reactions to vaccines [13,14]. PEG is implied in mRNA vaccines, whereas Polysorbate 80 is in adenoviral vector vaccines. Tromethamine is present in the mRNA-1273 vaccine only. PEG and Polysorbate 80 share structural similarities, and a recent study showed that cross-reactive type 1 hypersensitivity to PEG and Polysorbate 80 may occur [15].

Moreover, those with evidence of an allergic reaction following the first dose of any COVID-19 vaccine are advised not to receive a second dose, and they should refer to an allergist [16]. 

In order to complete the vaccination in patients with a suspect allergic reaction following the first dose of any anti-SARS-CoV-2 vaccine and in those with evidence of sensitization to any of the allergenic excipients of the current available anti-SARS-CoV-2 vaccines, desensitization procedures may be helpful in optimizing the outcomes of the immunization campaign [4]. As stated in the European Academy of Allergy and Clinical Immunology (EAACI) position paper on rapid desensitization for drug hypersensitivity, drug desensitization is indicated when the culprit drug is irreplaceable or more effective than the alternative, or it has a unique mechanism of action. Desensitization procedures should be avoided in patients with uncontrolled asthma or uncontrolled cardiac diseases. Desensitization is completely contraindicated in patients who have experienced severe hypersensitivity reactions, such as Stevens–Johnson Syndrome, Toxic epidermal necrolysis (TEN), Drug-Induced Hypersensitivity Syndrome (DIHS) or Drug Reaction with Eosinophilia and Systemic Symptoms (DRESS) [17]. 

A single standardized scheme for desensitization to anti-SARS-CoV-2 vaccines has not been established; however, new promising approaches are being considered, and few publications reported case reports or small case series of patients receiving an anti-SARS-CoV-2 vaccine with a desensitization protocol [9,18,19,20,21,22,23].

Given the few publications on the standardization of graded dosing desensitization protocols with anti-SARS-CoV2 vaccines, we aimed to describe our experience with a larger group of patients compared to the currently published reports.

## 2. Materials and Methods

### 2.1. Study Design

This is a retrospective analysis of all consecutive patients who underwent administration of the SARS-CoV-2 vaccine with a desensitization protocol at our Center since the beginning of the vaccination campaign (January 2021) and until September 2022. All patients had had a suspected hypersensitivity reaction to a previous administration of the vaccine or were sensitized to at least one excipient of the anti-SARS-CoV-2 vaccines (in particular, PEG, Polysorbate 80 and Tromethamine). 

Delayed adverse reactions occurring later than 72 h after vaccination were excluded because of the difficulty in demonstrating a clear correlation to the vaccine. 

### 2.2. World Allergy Organization (WAO) Systemic Allergic Reaction Grading System

Hypersensitivity reactions have been classified in terms of severity as recommended by the World Allergy Organization (WAO) systemic allergic reaction grading system [24] that classifies allergic reactions into 5 grades based on organ system involved (cutaneous, conjunctival, upper respiratory, lower respiratory, gastrointestinal, cardiovascular, and other) and severity. A reaction from a single organ system such as cutaneous, conjunctival or upper respiratory, but not asthma, gastrointestinal or cardiovascular is classified as a grade 1. Symptoms/signs from more than 1 organ system or asthma, gastrointestinal, or cardiovascular are classified as grades 2 or 3. Respiratory failure or hypotension, with or without loss of consciousness, defines grade 4 and death grade 5. The grade is determined by the physician’s clinical judgment.

### 2.3. Desensitization Protocols

The desensitization protocols we developed consisted of the fractioned administration of the entire vaccine dose into 4 separate injections of increasing quantity (Table 1) through a 2-h period and followed by 2-h of observations, according to what was suggested by Kelso et al. [25] and following the general recommendations given by the European Network on Drug Allergy (ENDA) and EAACI position paper on allergies and COVID-19 vaccines [26] that suggests taking into account the different total volumes of the currently available vaccines (e.g., 0.3 mL of BNT162b2 and 0.5 mL of mRNA-1273 and ChAdOx1-S).

### 2.4. Brighton Collaboration Case Definition Criteria for Anaphylaxis

Adverse reactions to desensitization protocol were classified according to The Brighton Collaboration case definition criteria for anaphylaxis [7]. The Brighton classification has 5 levels: level 1 represents the highest level of diagnostic certainty that a reported case is indeed a case of anaphylaxis; levels 2 and 3 represent lower levels of diagnostic certainty. Level 4 describes a case of anaphylaxis that does not fulfill the Brighton Collaboration case definition criteria. Level 5 is a case that was neither reported as anaphylaxis nor meets the case definition.

All patients signed an informed consent for desensitization procedure and for publication of anonymized data.

## 3. Results

Thirty consecutive patients were included in this analysis: 27 (90.0%) had a previous immediate hypersensitivity reaction to an anti-SARS-CoV-2 vaccine (24 after the first dose and 3 after the second one): 19 out of 27 (70.4%) grade 1, 5 (18.5%) grade 2, and 3 (11.1%) grade 3 according to the WAO criteria for systemic reaction severity [24]. Three patients (10.0%) had positive skin test for at least one vaccine excipient (PEG or Polysorbate-80) and a clinical history of suspected allergic reactions to any drug containing those excipients.

Among 27 patients that underwent desensitization because of hypersensitivity reactions after anti-SARS-CoV-2 vaccine administration, 17 (63.0%) experienced a reaction following vaccination with BNT162b2, 6 (22.2%) with mRNA-1273 and 4 (14.8%) with ChAdOx1-S. (Table 2). The vaccine that was associated with the hypersensitivity reaction was preferentially chosen to continue the immunization process through a desensitization procedure; in 13 cases, however, for reasons independent of the authors of this article and the health professionals who conducted the desensitization procedure, and mainly due to which products were available and distributed by the health authorities to our vaccination center, the choice of vaccine used for desensitization differed from that associated with the hypersensitivity reaction.

Twenty-eight patients (93.3%) had no hypersensitivity reactions during and after vaccination administered with desensitization protocol. One patient (patient n.3 in Table 2) experienced diffuse itching and significant hypotension (Brighton 1) during desensitization with the BNT162b2 vaccine. The patient was immediately treated with 500 mcg of intramuscular epinephrine, and after resolution, within a few minutes, the patient was nonetheless transferred to our Emergency Department for observation. Prior to the vaccination following the desensitization protocol, this patient underwent skin prick testing for both Polysorbate-80 and PEG, and interestingly only dermographism was reported. Another patient (n.23 in Table 2) experienced diffuse itching, angioedema, and nausea (Brighton 3) during desensitization with the mRNA-1273 vaccine. Symptoms completely resolved within one hour after intramuscular administration of antihistamine (Chlorpheniramine 10 mg) and corticosteroids (Beclometasone 4 mg). The patient was dismissed with allergological contraindication to the booster dose. In both cases, the desensitization protocol was interrupted. Among the 3 patients with a positive history of previous suspected allergic reactions to drugs containing vaccine excipients and positive skin tests to them (all of them resulted positive for Polysorbate 80), none developed hypersensitivity reactions during or after the desensitization protocol with BNT162b2 vaccine. 

No late hypersensitivity reactions to desensitization protocol were brought to our attention. 

## 4. Discussion

Here, we have described our experience of administering anti-SARS-CoV-2 vaccines using desensitization protocols in patients at high allergic risk. In our series, we desensitized patients with three different types of vaccines, using administration schedules specially developed by us since, at the time of the procedures, there was no validated and unanimously recognized protocol in the literature for desensitization with anti-SARS-CoV-2 vaccines. The desensitization protocol scheme for the BNT162b2 vaccine used in our series corresponds to the one we previously published in the context of an article in which we described the risk management protocol implemented during the vaccination campaign of the healthcare personnel of our hospital, but on that occasion applied to a single patient [9]. In the present case series, the same desensitization protocol with the BNT162b2 vaccine was applied to 15 patients proving to be safe and able to lead to the completion of the vaccination in 14 patients. In a patient with a history of acute urticaria following the first dose of the SARS-CoV-2 vaccine and skin tests for vaccine excipients uninterpretable for demographics, the desensitization procedure was associated with anaphylaxis and was, therefore, ineffective. Recently, Iemoli et al. described two patients who experienced bronchospasm after vaccination with the BNT162b2 vaccine and for whom gradual administration of incremental doses of vaccine was not sufficient to prevent the recurrence of symptoms [21]. Other experiences, on the other hand, are reassuring of the efficacy of desensitization protocols with the BNT162b2 vaccine [22,23]. As regards the other vaccines (mRNA-1273 and ChAdOx1-S), the desensitization protocols used were designed according to the international recommendations for desensitization to drugs and vaccines [9,25,26], proving to be safe and effective.

In two patients, the desensitization procedure was associated with systemic hypersensitivity reactions; this partly falls within the expected occurrences as no desensitization procedure is expected to have 100% efficacy. The two patients were subsequently also studied for clinical conditions predisposing to more serious allergic reactions, such as systemic mastocytosis, with negative results. No other known predisposing factors were found in these two patients. Therefore, we can only assume that those two patients have a lower intrinsic reactivity threshold for the allergenic stimuli of the SARS-CoV-2 vaccine. It is theoretically possible that desensitization schemes starting with lower vaccine doses and with smaller gradual dose increments may be better tolerated by these two patients; unfortunately, we have no way of confirming this hypothesis as the two patients refused to continue with the vaccination after the reaction occurred during desensitization.

In addition to the immunological efficacy of vaccine desensitization procedures, they may also help overcome the reasonable concerns of patients who have experienced hypersensitivity reactions to a previous dose of the vaccine or who have a clinical history and allergy tests confirming allergy to at least one of the excipients of the vaccines. In this sense, desensitization protocols should be taken into consideration and become one of the procedures routinely offered to patients at high risk of allergic reactions to SARS-CoV-2 vaccines. 

It could be argued that the graduated administration of the vaccine may not be able to provide similar protection against infection; however, the total dose administered with the desensitization protocol does not differ from that administered in a single shot, and to date, there are no data that can reasonably raise concerns about a lower efficacy of a graduated vaccination compared to the usual one. Interestingly, recently Regula et al. demonstrated that graded-dose administration of whole-body anti-SARS-CoV-2 vaccines was associated with a significant increase in the serum concentration of SARS-CoV-2 spike IgG [18], confirming that desensitization protocols are also able to provide appropriate protection immunology for the patient.

Furthermore, it is known that a single dose of the SARS-CoV-2 vaccine is not able to provide adequate immunological protection [27], and therefore allows patients who experienced possibly allergic reactions after the first dose to complete the vaccination series (through desensitization) to become dutiful and ethically relevant action both in terms of individual protection and the wider effect on public health; related to this, most of our patients (except the two who developed anaphylaxis during the desensitization procedure) managed to receive the subsequent doses of vaccine always applying the desensitization protocol.

A possible alternative option to vaccine desensitization could be to desensitize patients directly with excipients suspected of being the relevant allergen in hypersensitivity reactions to SARS-CoV-2 vaccines (in particular, PEG and Polysorbate 80). To date, the scientific literature reports that this option has only been explored in only two single cases: a patient with certified PEG allergy underwent a PEG desensitization protocol followed by an already scheduled colonoscopy (PEG is used in preparation for the colonoscopy) and anti-SARS-CoV-2 vaccination with BNT162b2 [28]; in another case, a patient with anaphylaxis after preparation for PEG colonoscopy, found to be sensitized to both PEG and polysorbate 80 and scheduled to start anti-Tumor Necrosis Factor (TNF) alpha drug therapy for Crohn's disease, was successfully desensitized with infliximab (an anti-TNF alpha drug with polysorbate 80 as excipient) and subsequently vaccinated against SARS-CoV-2, without hypersensitivity reactions, with Ad26.COV2.S vaccine containing polysorbate 80 as excipient [29]. We did not consider it appropriate to desensitize with PEG or Polysorbate 80, as the role of these excipients in determining hypersensitivity reactions to anti-SARS-CoV-2 vaccines is still not fully understood [13]. Instead, we preferred to use a more pragmatic and direct approach to the product that caused the reactions, choosing to desensitize with the entire vaccine.

## 5. Conclusions

In conclusion, in this article, we have presented our series, to date the largest published so far, of desensitization to anti-SARS-CoV-2 vaccines, also proposing specific protocols that have proved to be safe and able to allow the completion of the vaccination even in patients with previous allergic reactions to the same products.

## Figures and Tables

**Table 1 vaccines-11-00910-t001:** Desensitization protocols for each SARS-CoV-2 vaccine.

		BNT162b2 (Pfizer BioNTech)	mRNA-1273 (Moderna)	mRNA-1273 (Moderna)—Booster Dose	ChAdOx1-S (Astra Zeneca)
Step	Time (min)	Administered Dose (Cumulative Dose)	Administered Dose (Cumulative Dose)	Administered Dose (Cumulative Dose)	Administered Dose (Cumulative Dose)
1	0	0.03 mL (0.03)	0.05 mL (0.05)	0.05 mL (0.05)	0.05 mL (0.05)
2	30	0.07 mL (0.10)	0.10 mL (0.15)	0.05 mL (0.10)	0.10 mL (0.15)
3	60	0.10 mL (0.20)	0.15 mL (0.30)	0.07 mL (0.17)	0.15 mL (0.30)
4	90	0.10 mL (0.30)	0.20 mL (0.50)	0.08 mL (0.25)	0.20 mL (0.50)

**Table 2 vaccines-11-00910-t002:** Summary of hypersensitivity reactions and desensitization protocol outcomes in the evaluated patients.

Patient N°	Age and Sex	Type and Dose of mRNA Vaccine with Standard Protocol		Reactions to Standard Protocol	WAO Grade of Systemic Allergic Reaction to Standard Protocol	Management of Adverse Reaction	Type and Dose of mRNA Vaccine with Desensitization Protocol			Reactions to Desensitization Protocol	Reason for Undergoing Desensitization Protocol
		1°	2°				1°	2°	3°		
1	F, 57	N/A	N/A	N/A	N/A	N/A	P	P	N/A	None	Positive skin test for Polysorbate-80
2	F, 33	N/A	N/A	N/A	N/A	N/A	P	P	N/A	None	Positive skin test for Polysorbate-80
3	F, 47	P	N/A	Urticaria	1	N/A	N/A	P	N/A	Anaphylaxis	Adverse reaction with standard protocol/non interpretable skin test to PEG and Polysorbate-80 for dermographism
4	F, 13	P	N/A	Urticaria (3 h)	1	Antihistamine and systemic corticosteroid	N/A	P	N/A	None	Adverse reaction with standard protocol
5	F, 34	P	N/A	Immediate lingual edema	3	Antihistamine and systemic corticosteroid	N/A	P	N/A	None	Adverse reaction with standard protocol
6	F, 47	M	N/A	Immediate rash	1	Antihistamine	N/A	M	M	None	Adverse reaction with standard protocol
7	F, 61	A	N/A	Urticaria	1	N/A	N/A	A	N/A	None	Adverse reaction with standard protocol
8	F, 62	A	N/A	Immediate pruritus, neck erythema, and laryngeal constriction	2	Antihistamine and systemic corticosteroids	N/A	A	M	None	Adverse reaction with standard protocol
9	F, 59	P	N/A	Immediate rash and pruritus	1	Antihistamine	N/A	P	M	None	Adverse reaction with standard protocol
10	F, 70	P	N/A	Urticaria	1	Not reported	N/A	P	M	None	Adverse reaction with standard protocol
11	F, 83	P	N/A	Immediate nose pruritus and glottic edema	3	N/A	N/A	P	P	None	Adverse reaction with standard protocol
12	F, 49	P	N/A	Diffuse pruritus	1	Antihistamine	N/A	P	M	None	Adverse reaction with standard protocol
13	F, 31	P	N/A	Pruritic papular rash	1	No treatment	N/A	P	N/A	None	Adverse reaction with standard protocol
14	F, 59	N/A	N/A	N/A	N/A	N/A	P	P	P	None	Positive skin test for Polysorbate-80
15	F, 25	P	N/A	Angioedema	1	Antihistamine and systemic corticosteroids	N/A	P	N/A	None	Adverse reaction with standard protocol
16	F, 53	P	N/A	Angioedema	1	N/A	N/A	P	N/A	None	Adverse reaction with standard protocol
**17**	F, 83	P	N/A	Immediate erythema of the hands and lingual edema	1	Antihistamine	N/A	P	M	None	Adverse reaction with standard protocol
**18**	F, 61	A	N/A	Immediate pruritic rash of the trunk	1	Antihistamine and systemic corticosteroids	N/A	A	M	None	Adverse reaction with standard protocol
**19**	M, 71	A	N/A	Angioedema, dyspnea, and malaise	2	N/A	N/A	P	P	None	Adverse reaction with standard protocol
**20**	F, 28	M	N/A	Immediate diffuse pruritic rash	1	Antihistamine	N/A	N/A	M	None	Adverse reaction with standard protocol
**21**	F, 55	P	N/A	Immediate dyspnea, sore throat, dysphagia, and delayed cutaneous rash	2	Antihistamine	N/A	M	N/A	None	Adverse reaction with standard protocol
**22**	F, 34	M	N/A	Pruritus sine materia	1	Systemic corticosteroids	N/A	M	N/A	None	Adverse reaction with standard protocol
**23**	F, 48	M	N/A	Immediate malaise, erythematous rash, and pruritus	1	Systemic corticosteroids	N/A	M	N/A	Pruritus, erythematous rash, angioedema, nausea	Adverse reaction with standard protocol
**24**	M, 28	P	N/A	Immediate erythematous rash	1	Antihistamine	N/A	M	N/A	None	Adverse reaction with standard protocol
**25**	F, 45	M	N/A	Immediate pruritus and palpebral edema	1	N/A	N/A	M	N/A	None	Adverse reaction with standard protocol
**26**	F, 42	M	N/A	Angioedema and erythematous rash	1	Antihistamine	N/A	M	M	None	Adverse reaction with standard protocol
**27**	F, 32	P	P	Glottic edema and dysphagia	2	N/A	N/A	N/A	M	None	Adverse reaction with standard protocol
**28**	F, 41	P	N/A	Hypotension and cephalgia	2	Anti-inflammatory	N/A	M	N/A	None	Adverse reaction with standard protocol
**29**	F, 57	P	P	Immediate rash	1	N/A	N/A	N/A	M	None	Adverse reaction with standard protocol
**30**	F, 60	P	P	Immediate dysphagia, cough, and pruritic, erythematous palmar rash	3	N/A	N/A	N/A	M	None	Adverse reaction with standard protocol

A: ChAdOx1-S (AstraZeneca), M: mRNA-1273 (Moderna), P: BNT162b2 (Pfizer), PEG: Polyethylene glycol, WAO: World Allergy Organization.

## Data Availability

The data presented in this study are openly available in Zenodo at this link: https://zenodo.org/communities/humanitasirccs/.

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
