# Peer review of "Desensitization Protocols for Anti-SARS-CoV-2 Vaccines in Patients with High Risk of Allergic Reactions"

_vaccines, 2023, doi:10.3390/vaccines11050910_

Round 1

Reviewer 1 Report

I consider your work very important for all physicians. Compliments

Author Response

Thank you very much for appreciating our article

Reviewer 2 Report

The group's work adds important knowledge in cases of individuals who have an allergic reaction to the components of anti-COVID-19 vaccines, but some points were confused

1 - After the desensitization protocol, did the individuals complete the protocol receive the second dose of the vaccine?

2 - Since the desensitization protocol used complete vaccines (containing anti-SARSCOV antigens) was the specific antibody response evaluated in these individuals? I believe that the desensitization protocol using the whole vaccine may alter the vaccine-induced antibody response in individuals desensitized with the protocol described by the authors, compared to the response seen in non-sensitized individuals. Without these data, it would not be possible to release this protocol, since it would not be appropriate to use a protocol that, despite desensitizing against Polyethylene glycol (PEG), Polysorbate 80 and Tromethamine, interferes with the specific immune response to the infectious agent.

Author Response

Dear Reviewer,

Thank you very much for appreciating our article and for your valuable comments.

Please find here below our point-by-point response to your comments:

1 - After the desensitization protocol, did the individuals complete the protocol receive the second dose of the vaccine?

RESPONSE: Yes, apart from the two patients who had to interrupt the desensitization procedure due to the onset of anaphylaxis, all the others completed the vaccination cycle, always by means of desensitization protocols. We elaborated on this in the Discussion of the revised article.

2 - Since the desensitization protocol used complete vaccines (containing anti-SARSCOV antigens) was the specific antibody response evaluated in these individuals? I believe that the desensitization protocol using the whole vaccine may alter the vaccine-induced antibody response in individuals desensitized with the protocol described by the authors, compared to the response seen in non-sensitized individuals. Without these data, it would not be possible to release this protocol, since it would not be appropriate to use a protocol that, despite desensitizing against Polyethylene glycol (PEG), Polysorbate 80 and Tromethamine, interferes with the specific immune response to the infectious agent.

RESPONSE: We thank the Reviewer for his/her comments. However, we must point out that graded-dose administration (desensitization) approaches with the whole vaccine are suggested by all official documents (e.g.: CDC Vaccine Recommendations and Guidelines of the Advisory Committee on Immunization Practices, ACIP; EAACI position paper on vaccination and allergy; EAACI statement on the diagnosis, management and prevention of severe allergic reactions to COVID-19 vaccines...) to allow the completion of vaccination in patients at high risk of allergies following the administration of a vaccine, including those for SARS-CoV -2. There are few direct confirmations in the literature of obtaining protection from infection following desensitization with a whole vaccine, but some data has been published precisely for anti-SARS-CoV-2 vaccines (Regula P et al. - J Allergy Clin Immunol Glob. 2022 Aug; 1(3): 175–177.). We already acknowledged this aspects in the Discussion of the original version of our manuscript, and we added other consideration related to these aspects in the Discussion of the revised article.

Reviewer 3 Report

1.This is a short report on the experience of Desensitization protocols for anti-SARS-CoV-2 vaccines in patients with high risk of allergic reactions, which has good reference value.

2.Suggest changing abstract(Line 21-23)“We here describe our experience on 30 patients with previous allergic reactions to anti-SARS-CoV-2 vaccines or to any of their excipients (so far the largest case series published), proving that they are effective and safe: only two patients experienced hypersensitivity reaction symptoms during the desensitization procedure. toWe here describe our experience on 30 patients with previous allergic reactions to anti-SARS-CoV-2 vaccines or to any of their excipients, proving that they are effective and safe: only two patients experienced hypersensitivity reaction symptoms during the desensitization procedure. 

3.It is recommended to introduce the design basis of the desensitization protocols in the materials 。

4.1. ï¼ˆLine141-144)“For 14 of these patients the desensitization procedure was performed using the same vaccine that elicited hypersensitivity reaction, while for the remaining 13 patients the choice of the vaccines used for the desensitization procedure was conditioned by their availability.The author needs to explain why the 13 patients need to choose different vaccines, and how to choose the vaccine by their availability? 

5.The author needs to discuss the possible causes of allergic reactions in 2 patients and propose new desensitization protocols.

6.Proof of review by the ethics committee is required.

Author Response

Dear Reviewer,

Thank you very much for appreciating our article and for your valuable comments.

Please find here below our point-by-point response to your comments:

1. This is a short report on the experience of Desensitization protocols for anti-SARS-CoV-2 vaccines in patients with high risk of allergic reactions, which has good reference value.

RESPONSE: Thank you again for the appreciation

2. Suggest changing abstract(Line 21-23)“We here describe our experience on 30 patients with previous allergic reactions to anti-SARS-CoV-2 vaccines or to any of their excipients (so far the largest case series published), proving that they are effective and safe: only two patients experienced hypersensitivity reaction symptoms during the desensitization procedure. ”to“We here describe our experience on 30 patients with previous allergic reactions to anti-SARS-CoV-2 vaccines or to any of their excipients, proving that they are effective and safe: only two patients experienced hypersensitivity reaction symptoms during the desensitization procedure. ”

RESPONSE: Thanks for the suggestion that we followed in the revised Abstract.

3. It is recommended to introduce the design basis of the desensitization protocols in the materials 。

RESPONSE: The desensitization protocols are already described in details in the sub-section 2.3 of the Methods. We added additional information on the references used to develop the desensitization protocols

4. (Line141-144)“For 14 of these patients the desensitization procedure was performed using the same vaccine that elicited hypersensitivity reaction, while for the remaining 13 patients the choice of the vaccines used for the desensitization procedure was conditioned by their availability.” The author needs to explain why the 13 patients need to choose different vaccines, and how to choose the vaccine by their availability? 

RESPONSE: thank you for this relevant comment. We re-formulated (and hopefully better explained) the sentence in which we commented this (see Results) into: "The vaccine that was associated with the hypersensitivity reaction was preferentially chosen to continue the immunization process through a desensitization procedure; in 13 cases, however, for reasons independent of the authors of this article and the health professionals who conducted the desensitization procedure, and mainly due to which products were available and distributed by the health authorities to our vaccination center, the choice of vaccine used for desensitization differed from that associated with the hypersensitivity reaction."

5. The author needs to discuss the possible causes of allergic reactions in 2 patients and propose new desensitization protocols.

RESPONSE: We added a paragraph in the Discussion commenting and exploring this very relevant point.

6. Proof of review by the ethics committee is required.

RESPONSE: as stated in the appropriate section at the end of the manuscript, "Ethical review and approval were waived for this study due to the pure descriptive nature of common clinical practice (patients signed an informed consent for the desensitization procedure, as per good clinical practice, and for publication of anonymized data)."